*GigaScience*, 2024, **13**, 1–10

**Tech Note**

# CoCoPyE: feature engineering for learning and prediction of genome quality indices

Niklas Birth [iD][†], Nicolina Leppich[†], Julia Schirmacher [iD], Nina Andreae, Rasmus Steinkamp, Matthias Blanke [iD], and Peter Meinicke [iD]*

Department of Applied Bioinformatics, Institute of Microbiology and Genetics, University of Goettingen, Goldschmidtstr. 1, 37077 Goettingen, Germany
*Correspondence address. Peter Meinicke, Department of Applied Bioinformatics, Institute of Microbiology and Genetics, University of Goettingen, Goldschmidtstr. 1, 37077 Goettingen, Germany. E-mail: pmeinic@gwdg.de
†Contributed equally.

## Abstract

**Background:** The exploration of the microbial world has been greatly advanced by the reconstruction of genomes from metagenomic sequence data. However, the rapidly increasing number of metagenome-assembled genomes has also resulted in a wide variation in data quality. It is therefore essential to quantify the achieved completeness and possible contamination of a reconstructed genome before it is used in subsequent analyses. The classical approach for the estimation of quality indices solely relies on a relatively small number of universal single-copy genes. Recent tools try to extend the genomic coverage of estimates for an increased accuracy.
**Results:** We developed CoCoPyE, a fast tool based on a novel 2-stage feature extraction and transformation scheme. First, it identifies genomic markers and then refines the marker-based estimates with a machine learning approach. In our simulation studies, CoCoPyE showed a more accurate prediction of quality indices than the existing tools. While the CoCoPyE web server offers an easy way to try out the tool, the freely available Python implementation enables integration into existing genome reconstruction pipelines.
**Conclusions:** CoCoPyE provides a new approach to assess the quality of genome data. It complements and improves existing tools and may help researchers to better distinguish between low-quality draft and high-quality genome assemblies in metagenome sequencing projects.

**Keywords:** metagenomics, quality assessment, machine learning

## Introduction

The number of genomes assembled from metagenome sequencing projects increases rapidly. However, the variation in the quality of the reconstructed genomes is considerable [1]. For all subsequent analyses that make use of these data, information about the completeness and purity of these genomes is essential. Therefore, an important aspect is the quality assessment of the assembled genomes in terms of completeness and contamination estimates: how much of the original genome is actually present, and how much of the genomic material possibly stems from other organisms? It has been suggested that a high-quality genome assembly should be complete by more than 90% with a contamination below 5% [2]. According to the MIMAG, standard high-quality genomes also must contain transfer RNA and ribosomal RNA (SSU/LSU) genes.

The standard for measuring the quality indices of bacterial and archaeal genomes is to count the occurrences of universal single-copy marker genes in the query genomes. The evident problem of this classical approach is that these marker genes only represent a small part of the entire genome. Universal single-copy gene (SCG) sets for archaea and bacteria typically contain fewer than 50 genes, which can only cover a few percent of an average size microbial genome [3, 4]. As a consequence, estimation of genome quality indices can become highly unreliable. In particular, it is difficult if not impossible to distinguish between indicators for completeness and contamination. In fact, it is easy to mix up 2 different genomes to equal proportions in such a way that a high

completeness with almost no contamination may be predicted as long as the few marker genes from the 2 organisms complement each other.

CheckM [1] tries to overcome these problems by using lineage-specific SCG sets. If a candidate genome can be assigned to one of the clades represented in CheckM, the tool uses a specific SCG set, which may comprise more than a thousand genes, depending on the similarity and the number of reference genomes in the assigned clade. The assignment in CheckM is based on phylogenetic placement of universal SCGs. This considerably improves the completeness and contamination estimates in cases where closely related genomes are present in the CheckM reference tree. If an assignment to a more specific clade is not possible, CheckM automatically uses a universal SCG set for prediction. Although the identification of lineage-specific SCG sets can substantially enlarge the statistical basis for estimation of quality indices, the corresponding procedure to establish and maintain the SCG sets for all nodes of the reference tree is rather complex [1]. BUSCO [5] offers a similar approach that is also based on prior identification of lineage-specific markers. In contrast to these marker-based approaches, also novel methods have been introduced that utilize machine learning [6–8] to assess the quality of metagenome assembled genomes. In principle, these methods may also work for candidate genomes for which no closely related reference genomes exist in current data bases. One of these tools is the successor to CheckM, CheckM2 [6], which predicts genome quality indices directly from a high-dimensional genomic feature space.

With CoCoPyE, we have developed a hybrid approach for the estimation of genome quality indices that combines the concept of marker genes with a machine learning approach. The corresponding tool is available for offline installation under the main operating systems and can be accessed online by means of the CoCoPyE web server.

## Methods

### Hybrid approach to quality assessment

Our quality assessment approach is based on the protein domain profile of a query genome. This profile contains the genome-specific frequencies of all protein domain families according to the Pfam database [9]. Based on these counts, the prediction of completeness and contamination is achieved in 2 stages: in stage I, suitable genomes for comparative analysis are identified in a reference database by a profile similarity search. The dynamically extracted marker domains of these reference genomes serve as a basis to establish a first estimate of quality indices. If either the predicted completeness is below 60% or the predicted contamination is above 30%, the estimates are directly reported as final predictions. Otherwise, the estimates are further refined in stage II. Thereby, the high-dimensional profile space is transformed to a low-dimensional feature space based on all Pfam count ratios between query genomes and reference profiles without restriction to a specific set of marker domains. The corresponding features are finally used for a machine learning–based prediction. An important difference between the 2 stages is that stage I can work with a wide range of genome quality because it is not trained for a particular range. In contrast, stage II is subject to specific training with data from a defined quality range and therefore requires stage I for filtering and prediction in cases where the stage I estimates indicate that the stage II quality range is not met. An overview of the complete prediction engine is shown in Fig. 1.

### Protein domain features

The analysis of a query genome starts with a protein domain search with UProC [10] using default parameter values and counting the occurrences of protein sequence families within potential coding regions as obtained from all translated open reading frames with a minimum length of 20 amino acids found in the genomic sequences. This results in a high-dimensional profile of protein domain counts. As a protein database, we offer 2 preprocessed versions of Pfam [9], which in case of versions 24 and 28 result in 11,912 and 16,230 features, respectively.

### Prefiltering

A central step in our method is the search for nearest neighbors of a query genome in the reference database. This step becomes unreliable if the completeness of the input is too low (i.e., if the input lacks a sufficient number of potential protein domain markers). For this reason, we require an initial completeness estimate as obtained from 2 superkingdom-specific marker sets, according to all bacterial and archaeal reference genomes. For each set, we applied a 95% coverage criterion on single-copy domains to define the initial markers. If the completeness estimates with regard to both marker sets are below 10%, we reject the query.

### Nearest neighbor search

Otherwise, the high-dimensional profile vector of a query genome is compared with the precomputed profiles in a reference genome database. For this, similar reference profiles are identified by $K$-nearest neighbor search: for protein family indices $i$, $j$, $k$ and protein family counts $C_q^{(i)}$ (query) and $C_r^{(i)}$ (reference), the similarity measure

$$\text{sim}\left(\vec{C}_q, \vec{C}_r\right) = \frac{\left|\left\{i \mid \left(C_q^{(i)} = C_r^{(i)}\right) \wedge \left(C_q^{(i)} > 0\right)\right\}\right|}{\sqrt{\left|\left\{j \mid C_r^{(j)} > 0\right\}\right| \cdot \left|\left\{k \mid C_q^{(k)} > 0\right\}\right|}} \tag{1}$$

counts the number of coinciding nonzero counts in corresponding profile entries. The required equality implies that mainly small counts contribute to the similarity estimate. In prior studies, we found that larger counts usually introduce too much variation and unfavorably increase the impact of possible contaminants on the similarity measure.

### Marker-based estimate with nearest reference profiles

From the $K$ nearest neighbors according to the above profile similarity measure, we compute an initial marker-based estimate of the completeness and contamination indices. We do not use predefined static marker sets but instead obtain a set of specific markers $\mathcal{M}$ for each query that arises from feature dimensions with equal nonzero counts in all $K$ nearest neighbors. With index set $\mathcal{I}_K$ containing all reference indices of the neighbors, we obtain the marker set

$$\mathcal{M} = \left\{m \mid \forall i, j \in \mathcal{I}_K : \left(C_i^{(m)} = C_j^{(m)}\right) \wedge \left(C_i^{(m)} > 0\right)\right\}. \tag{2}$$

According to this definition, markers are not restricted to single-copy protein domains. With the query-specific markers, we then apply the standard estimation scheme. From $M$ specific markers in $\mathcal{M}$ with reference counts $C_r^{(m)}$, we get the stage I contamination (cont) and completeness (comp) estimates

$$\text{cont} = \frac{1}{M} \sum_{m \in \mathcal{M}} \left[\frac{C_q^{(m)}}{C_r^{(m)}} - 1\right]_+, \tag{3}$$

$$\text{comp} = \frac{1}{M} \sum_{m \in \mathcal{M}} \frac{C_q^{(m)}}{C_r^{(m)}} - \text{cont}, \tag{4}$$

where $[z]_+ = \max(z, 0)$. If these estimates are within the above-mentioned range (contamination below 30% and completeness above 60%), stage II is utilized to refine these estimates. Otherwise, the predictions of stage I are reported as the final result.

### Count ratio histograms

The original feature space as described above comprises more than 10,000 dimensions, which correspond to different protein domain families. Large-scale machine learning within such a high-dimensional space is burdensome. While neural networks, in principle, are well suited for training with large data sets, a high-dimensional input space slows down iterative training and increases the risk of overfitting. Therefore, we mapped the original profile space to a lower-dimensional histogram space. A count ratio histogram (CRH) arises from the comparison of a candidate profile with a reference profile in terms of the observed ratios between the corresponding protein domain counts. More specifically, we consider all ratios $C_q^{(i)}/C_r^{(i)}$ between counts from the query (nominator) and reference (denominator) genomes, where $i$ is the Pfam domain index. Note that all Pfam domain families with nonzero counts are included in the CRH computation, not just the markers that have been used for the initial prediction in stage I.

As bin centers for the CRH, we use the set of possible ratios between counts in a range $1 \ldots c_{\max}$. Thus, the integer variable $c_{\max}$ specifies the resolution of the histogram. For example, with $c_{\max} = 4$, we obtain the bin centers

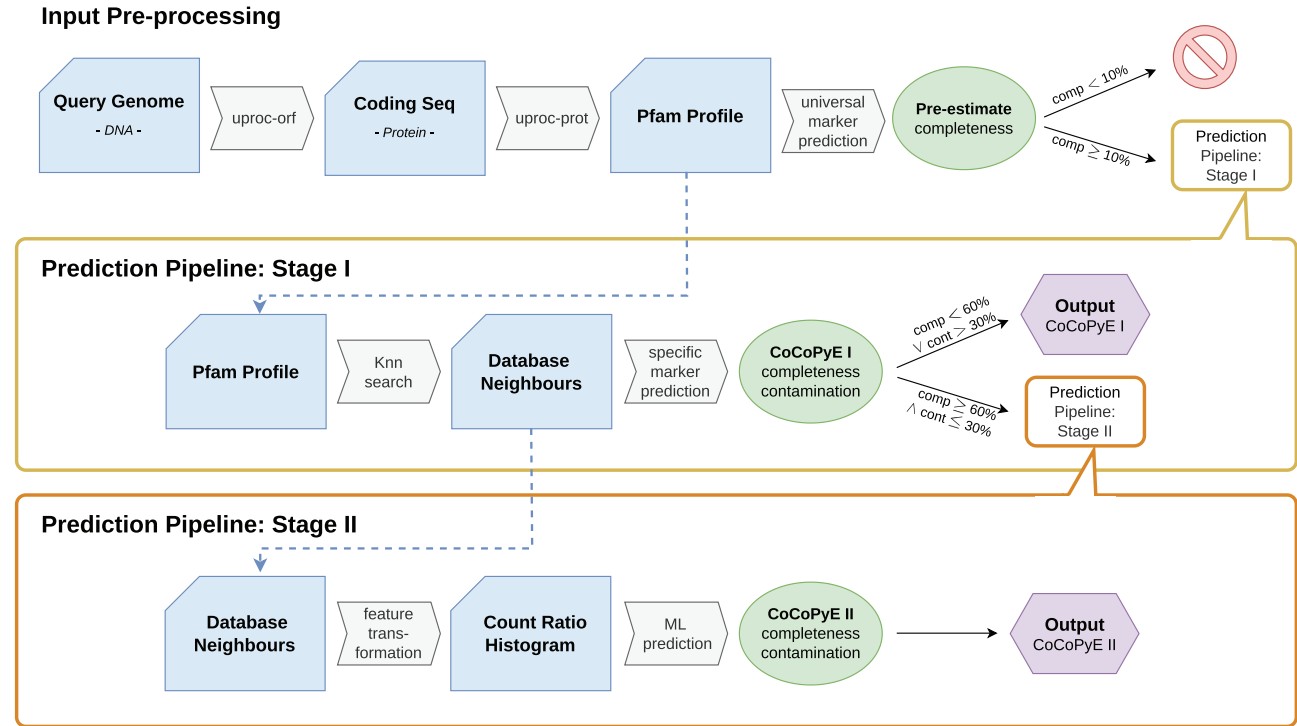

**Figure 1:** Schematic overview of the prediction pipeline.

$\mathcal{C} = \{1/4, 1/3, 1/2, 2/3, 3/4, 1, 4/3, 3/2, 2, 3, 4\}$. In addition, the left-most bin of the histogram also represents the domains of the reference genomes that have not been observed in the query genome. Similarly, on the other extreme of the histogram, we use the last right-most bin to also count all the protein families with a count ratio larger than $c_{max}$, and we also include those entries that occur in the query genome but not in the reference in this bin.

The histogram is normalized to relative frequencies. Therefore, a CRH with a single central peak at 1 would indicate a highly complete query genome without contamination. With a decreasing completeness, the variance of the count ratio distribution increases and more probability mass will be observed in noncentral bins on the left-hand side. In contrast, an increase of the right-hand side bins indicates a growing contamination. Two examples of CRHs for query genomes with different completeness and contamination values are shown in Fig. 2.

*Machine learning methods*

With the CRH feature vectors, we improved the marker-based estimates of the stage I prediction with different machine learning approaches in stage II. The query input of stage II was obtained from the average over all CRH vectors that result from comparison of the query with the $K$ nearest references in stage I. As additional features, we used the estimates of completeness and contamination as predicted in stage I. To identify suitable machine learning methods, we compared several linear and nonlinear regression techniques as implemented in `sklearn` Vers. 1.3.1 [11]. For linear prediction, we evaluated support vector machine (SVM) and elastic net regression, and as nonlinear approaches, we tested nearest neighbor, neural network, and random forest regression. For each method, we identified suitable values for the hyperparameters by performing a grid search for the stage I neighborhood size $K$ and for the CRH resolution $c_{max}$. Additionally, method-specific hy-

perparameters were included in the grid search where necessary. In particular, this included the SVM regularization parameter, the weight decay in the feed-forward neural networks with 1 hidden layer, and the smoothing parameter for the nearest neighbor regression. For the elastic net, we used the built-in hyperparameter optimization, and for random forests, we used the default parameter values in the `sklearn` implementation.

## Training and test data

*Reference database*

Our method requires a database of reference genomes for comparative analysis. As a basis, we use all genomes from the RefSeq [12] database with a *complete* or *chromosome* status annotation (download on 2 September 2023). We aim to provide references with a high quality that cover a wide range of different species without redundant protein profiles. We achieved this with a multistage filtering process: at first, we used UProC to determine the Pfam frequencies of all downloaded RefSeq genomes. We then applied an agglomerative clustering algorithm to the obtained frequencies in order to filter out closely related genomes, keeping only 1 representative for each cluster. Based on these representatives, we determined a set of single-count Pfam markers, separately for bacteria and archaea. We consider a protein family to provide such a general marker if and only if it occurs exactly once in at least 95% of our cluster representatives. This results in a set 107 markers for bacteria and 128 for archaea that we use to calculate a first completeness estimate of all downloaded genomes and removed those that had an estimated completeness of less than 95%. We applied this step to reduce the risk of including RefSeq entries with an erroneous completion state annotation. In a final step, all genomes that fulfilled the completeness criterion were clustered again, using the same method as before. The resulting cluster representatives constitute the genomes of our actual reference data set.

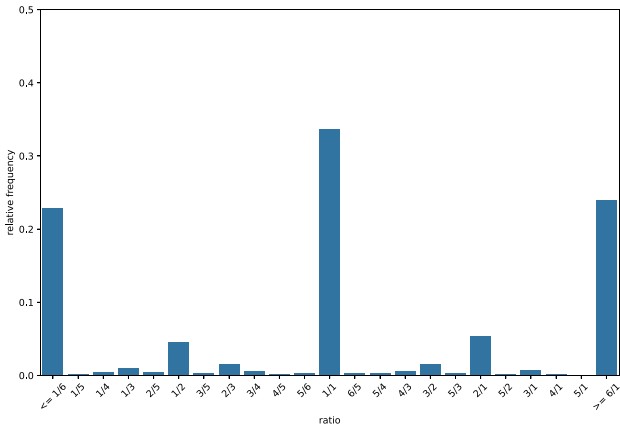 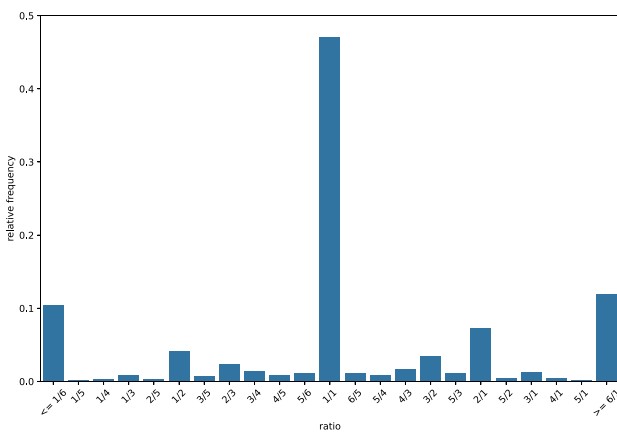

**Figure 2:** Count ratio histograms for 2 simulated example genomes with $c_{max} = 6$. The left histogram shows a bin with 61% completeness and 30% contamination. The right one shows a bin with 90% completeness and 10% contamination.

## Simulation

For our evaluation, we generated input query genomes with known ground truth according to specified values for completeness and contamination. In principle, we tried to simulate the results of a metagenome binning process that shows a broad spectrum of genome quality indices. First, all input genomes were fragmented, and fragments from different genomes were randomly combined to simulate variable degrees of completeness and contamination. For training, we used a 20-kb fragment length for simulated contigs. For the evaluation on separate test data, we also used other fragment lengths to study the impact of a differing contig length during prediction.

The simulation of a query genome was performed in the following way: first, random values for completeness and contamination were independently drawn from the ranges 60% to 100% and 0% to 30%, respectively. Then, a query genome was randomly selected from the set of candidate genomes, and fragments were randomly drawn to match the given completeness. Contamination refers to the original complete length of the query genome, and a corresponding number of fragments were drawn from a randomly selected contaminant genome. However, random selection is restricted to genomes with a similar genomic signature in terms of the similarity between tetramer frequency profiles. By this, we avoid mixing genomes with very different genomic signatures, which is usually not met in metagenomic binning results. If the selected contaminant genome is too short to provide the required contamination, more genomes are drawn from the similarity range until the specified contamination can be realized. For the range selection, we considered genomes according to tetramer profiles with a Bray–Curtis similarity ≥ 80 %.

## Training and validation

To obtain separate data sets for training of the machine learning methods and validation of the hyperparameters, we performed a data-splitting scheme based on the genome release date. The underlying idea is to simulate the novelty of genome data under realistic conditions, which imply a mixture of all kinds of evolutionary distances between the more recent genomes and previously published data. We divided the set of RefSeq cluster representatives into 3 subsets according to 2 split dates (5 May 2021 and 26 July 2022): the "oldest" and largest part 1 contains 6 036 genomes that served exclusively as reference genomes for the database nearest neighbor search of our approach. The middle part 2 and the

most recent part 3 were interchangeably used for training and validation, containing 1,503 genomes and 1,515 genomes, respectively. For each of the training/validation genomes, we simulated 20 bins with randomly chosen completeness and contamination (see above). The contaminant genomes were always drawn from the same part to avoid overlap between train and validation data. For part 2 and part 3, we realized a simple 2-fold cross-validation. In a single fold, 1 part was used for training and 1 for validation. The overall validation performance in terms of the mean absolute error (MAE) in percentage points was averaged over the 2 validation folds. Figure 3 shows an overview of the training/validation process. For training and validation, feature vectors were computed for all query bins generated from the corresponding parts. The training feature vectors were used to train the machine learning regression models for predicting completeness and contamination. The trained methods were then evaluated using the validation feature vectors.

According to the lowest validation error, we identified optimal hyperparameter values for the different machine learning approaches by grid search over a defined set of values. The ranges of the parameters can be found in Table 1. While some hyperparameters are specific to particular machine learning methods, the Pfam version, $K$ neighbors, and $c_{max}$ apply to all methods because they affect the input data for all learners.

## Test data and setup

After the selection of hyperparameters based on the 2-fold cross-validation, we chose the method with the best validation performance for the final prediction engine of CoCoPyE. In that way, we included 2 neural networks, 1 for the completeness and 1 for the contamination prediction. For the final version, the 2 networks were trained with both training sets (parts 2 and 3) combined, since there was no need for a separate validation fold anymore. Again, the first subset (part 1) was used to provide the reference database. After the training, the final reference database for CoCoPyE was built from all representatives of the RefSeq dataset (i.e., we extended the reference set by adding the genomes from both training sets).

We compared CoCoPyE with the existing tools CheckM (referred to as CheckM1 from now on) and CheckM2 in a comprehensive evaluation. We prepared 2 new genome sets A and B to ensure that the basis for the test data is distinct from that already used for the setup and training of the prediction engine.

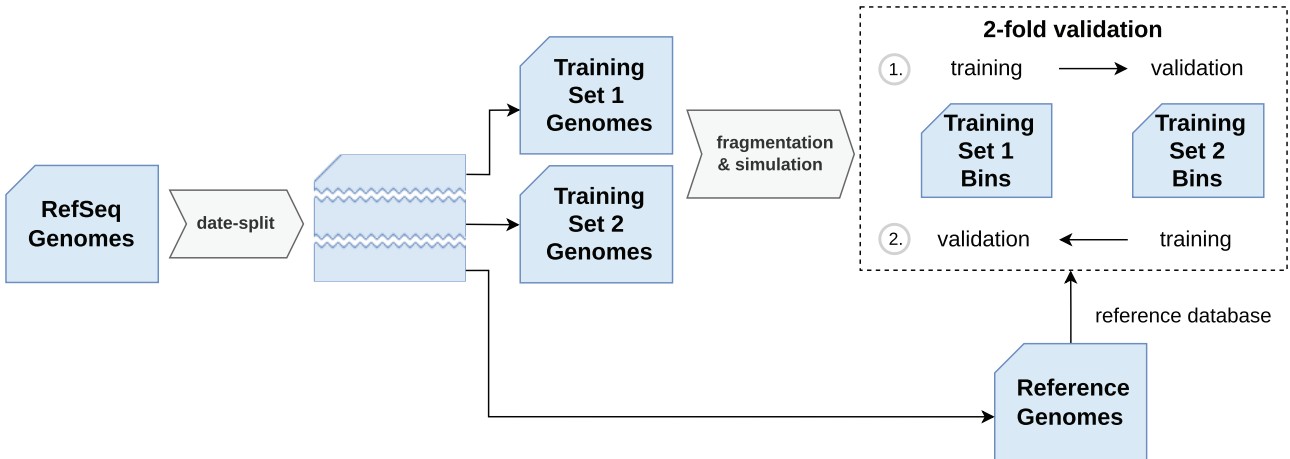

**Figure 3:** Schematic overview of the 2-fold training and validation process with 2 subsets (set 1, set 2). If the first subset is used for training, the second is used for validation and vice versa.

**Table 1:** Table of all possible hyperparameter ranges for the grid search

| Method | Hyperparameter | Value ranges |
|---|---|---|
| All | Pfam version | 24, 28 |
| All | K (ref. neighbours) | 1, 2, 3, 4, 5, 6, 7, 8, 9, 10 |
| All | $c_{max}$ | 4, 5, 6, 7, 8, 10, 12 |
| Linear SVR | C | 0.01, 0.1, 0.3, 0.5, 0.8, 0.9 |
| | | 1, 2, 3, 5, 10, 20, 40, 100 |
| | | 125, 150, 175, 200 |
| KNN regression | knn | 5, 10, 15, 20, 25, 30, 40, |
| | | 50, 60, 70, 80, 90, 100 |
| MLP | alpha | $10^{-4}$, $10^{-5}$, $10^{-6}$, $10^{-7}$ |

## Set A

From a phylogenomics study [13] that included a broad range of bacterial and archaeal genomes, we selected a subset of genomes according to assembly levels *contig* and *scaffold*, contrary to the selection criterion for our RefSeq data set. With the resulting 8,309 test genome candidates, we generated the test data. An overview of this process is shown in Fig. 4.

First, the genomes were analyzed by CheckM1, CheckM2, and CoCoPyE to provide a subset of test genomes where all 3 tools predict at least a completeness of 95% and a contamination of no more than 5%. The reason for this selection was 2-fold: first, we are mainly interested in the ability of the prediction engines to cope with varying degrees of incompleteness and contamination. In particular, the case where incomplete genomes are mixed with a significant contamination is an algorithmic and statistical challenge even if a method can recognize the uncontaminated complete versions as high-quality genomes. Furthermore, to account for the differing training and reference data used for implementation of the tools, it is fair to choose only test genomes for which all tools agree in the prediction of a high quality. Finally, we think this consensus analysis is the best way to cope with a missing ground truth, because for most of the reconstructed genomes in current databases, the actual quality of the reconstruction is not entirely clear. From the consensus analysis, we obtained 4,057 genomes, and Fig. 5 shows the agreement of the different tools. To avoid any direct overlap with our previous RefSeq-based data, we only considered genomes with a taxID not included in our training and

reference sets. After consensus analysis and taxID filtering, we finally obtained a test set of 3,540 genomes [14].

With these test genomes, we generated 3 separate sets of query bins. We applied the simulation scheme as described above to provide test sets with a 20-kb, 50-kb, and 100-kb fragment length. For each set, we generated 10 bins per test genome, again with randomly selected completeness and contamination values, so each of the 3 final test sets contained 35,400 bins [15]. The 3 tools were then run on all test data to evaluate their prediction performance.

## Set B

From the "Genomes from Earth's Microbiomes" (GEM) catalog [16], a collection of metagenome assembled genomes (MAGs), an additional test data set was built to study the prediction performance on simulated bins with real contig length distributions. Thus, we did not apply any fragmentation and used the sequences in terms of contigs as obtained from GEM. GEM contains 52,515 MAGs, of which we used 9,143 high-quality MAGs according to the MIMAG standard. Besides covering a wide range of microbial genomic diversity, the dataset provides information about habitats of the genetic material. This enabled us to conduct a more detailed performance analysis on environmental (further subdivided into aquatic and terrestrial), host-associated, and engineered MAGs. For the selection of suitable MAGs sufficiently distant from the references, we only considered test genomes with a minimum dissimilarity (Bray–Curtis) of 10% of the corresponding protein domain profile to the closest reference profile. In the same way as for the selection of our first test set, all compared tools must agree on the high quality of the full MAG sequence. Thus, we only chose those MAGs for testing for which all tools predicted a completeness ≥ 95% and a contamination ≤ 5%. Finally, to ensure variety in the test set, we clustered the remaining MAG protein domain profiles retaining 1 representative per cluster, similar to the reference set clustering before. This resulted in 1,210 test MAGs (421 environmental [thereof 297 aquatic, 124 terrestrial], 274 engineered, 515 host-associated) containing contigs, which were directly used for simulating test bins with different completeness and contamination values. The test bins were simulated in the same way as for test set A, except for the exclusion of the fragmentation step. For data set B, we finally obtained 12,004 test bins [15].

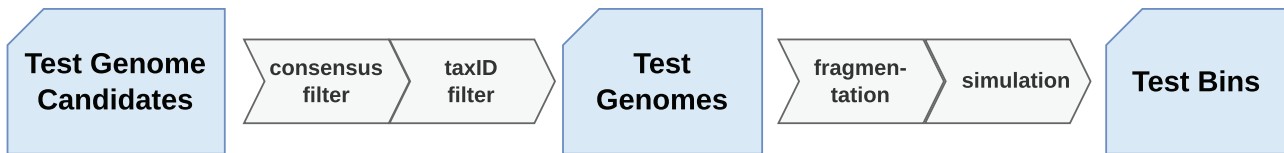

**Figure 4:** Schematic overview of the test data generation.

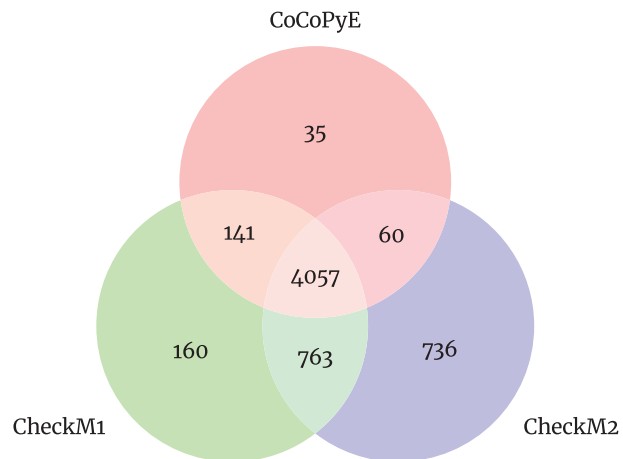

**Figure 5:** Number of test genome candidates that have at least a completeness of 95% and a contamination of 5% or less for CheckM1, CheckM2, and CoCoPyE.

**Table 2:** Mean and median absolute errors in percentage points of machine learning methods and the stage I prediction of CoCoPyE. Results on validation data are shown for the best-performing choice of hyperparameters for each method.

| Tool | Completeness error | | Contamination error | |
|---|---|---|---|---|
| | Mean | Median | Mean | Median |
| CoCoPyE stage I | 3.74 (±3.33) | 2.83 | 5.32 (±4.64) | 4.08 |
| ElasticNetCV | 2.97 (±2.64) | 2.28 | 4.24 (±3.24) | 3.55 |
| LinearSVR | 2.94 (±2.59) | 2.26 | 4.09 (±3.19) | 3.38 |
| KNN regression | 2.73 (±2.45) | 2.07 | 3.80 (±3.05) | 3.06 |
| Random forest | 2.63 (±2.41) | 1.98 | 3.68 (±3.01) | 2.95 |
| MLP | 2.58 (±2.38) | 1.95 | 3.50 (±2.94) | 2.77 |

## Results

### Validation and model selection

As described in our training and validation setup, we compared several machine learning models for stage II of our prediction engine and selected suitable values for the hyperparameters. Table 2 shows the mean and median absolute error in percentage points for the prediction of completeness and contamination on validation data. For comparison, we also included the results of the marker-based prediction of stage I. While the stage I prediction already works quite well, all methods in stage II could improve the initial results. Here, the nonlinear methods (K nearest neighbor [KNN], random forest [RF], multilayer perceptron [MLP]) show a slightly better prediction than the linear approaches (elastic net, support vector regression [SVR]). The best result was achieved by the neural network with an MAE of 2.58%pt and 3.50%pt for completeness and contamination, respectively. In this case, we used networks with 100 neurons and feature vectors based on

**Table 3:** Mean and median absolute error in percentage points for the test set A with 20-kb fragments

| Tool | Completeness error | | Contamination error | |
|---|---|---|---|---|
| | Mean | Median | Mean | Median |
| CoCoPyE | 3.09 (±3.05) | 2.26 | 4.49 (±4.52) | 3.29 |
| CheckM1 | 4.51 (±4.89) | 2.96 | 7.78 (±8.33) | 5.39 |
| CheckM2 | 6.13 (±5.67) | 4.45 | 7.39 (±6.20) | 5.78 |

Pfam version 28 and $K = 9$ neighbors. The network for the completeness prediction had the parameters $\alpha = 0.0001$, $c_{max} = 6$, and $\alpha = 0.0000001$ and $c_{max} = 12$ for the CRH and contamination. Both neural networks were included in the final version of the CoCoPyE prediction engine. The best result was achieved with the larger UProC database (Pfam 28) for protein domain detection. With the smaller database (Pfam 24), the MLP produced the best results, but the performance decreased slightly to 2.76%pt and 3.77%pt. This could justify the use of the smaller version in computers with limited RAM.

Comparing the different machine learning approaches, the performance differences between different methods were relatively small. Remarkably, RF regression with built-in default settings was very close to the best MLP results, although we did not perform any hyperparameter optimization.

### Performance comparison

In our evaluation of the final tool on test data, we compared CoCoPyE with CheckM1 [1] and CheckM2 [6]. For test data set A, the results in Table 3 indicate that CoCoPyE yields the lowest prediction error for both completeness and contamination, with an MAE of 3.09%pt and 4.49%pt. While CheckM1 outperforms CheckM2 in terms of a lower completeness error (4.51%pt vs. 6.13%pt), for the prediction of contamination, the MAE for CheckM1 is higher than for CheckM2 (7.78%pt vs. 7.39%pt).

For further analysis of the prediction error, we inspected the distributions of signed deviations, as shown in Fig. 6. While the completeness error of CoCoPyE shows a roughly symmetric distribution, CheckM2 shows a skewed distribution, indicating a clear tendency for overprediction of the quality index. A slight overprediction of completeness is also visible for CheckM1 and CoCoPyE, where CoCoPyE shows the lowest bias of all tools. The prediction of contamination also shows clear differences between the tools: while CoCoPyE yields an almost unbiased distribution, CheckM2 has a clear tendency to underestimate contamination. This is also visible in a weaker form for CheckM1, which, however, shows a highly asymmetric distribution with a heavy tail for the positive error.

We also investigated the performance for different fragment lengths of the simulated test data. Here, we observed a slight decrease in the prediction performance for all tools, as shown in Supplementary Tables S1 and S2 for 50-kb and 100-kb fragment

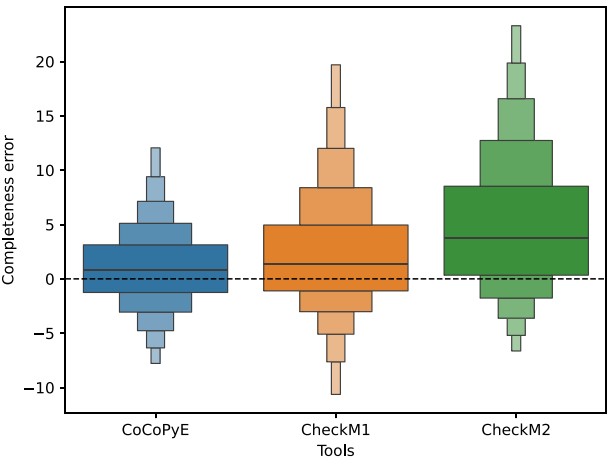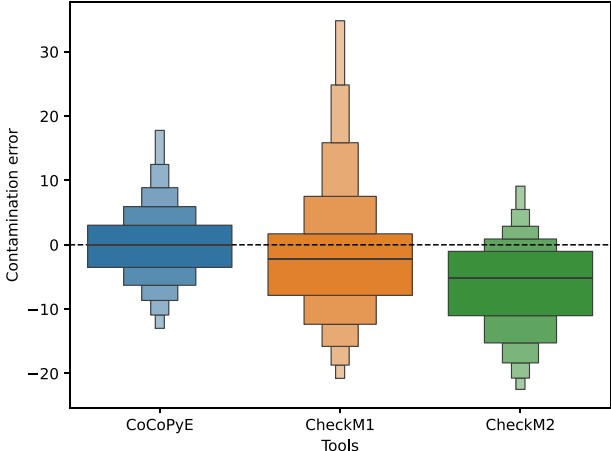

**Figure 6:** Signed error distributions of completeness (left) and contamination (right) for the compared tools on test set A with errors measured in percentage points. Note that the scale of the y-axis differs between the 2 plots.

**Table 4:** Mean and median absolute error in percentage points for test set B

| Tool | Completeness error | | Contamination error | |
|---|---|---|---|---|
| | Mean | Median | Mean | Median |
| CoCoPyE | 4.07 (±3.47) | 3.22 | 3.75 (±3.52) | 2.81 |
| CheckM1 | 5.21 (±5.17) | 3.68 | 5.85 (±6.18) | 3.95 |
| CheckM2 | 5.91 (±5.38) | 4.35 | 6.27 (±5.74) | 4.79 |

lengths. Because CoCoPyE was trained with 20-kb fragments, this result was not entirely surprising. Therefore, it could be beneficial to provide different neural networks trained with different fragment lengths. In this case, for the prediction at stage II, the network that best matches the average contig length of the input would be selected automatically.

To investigate if the CoCoPyE prediction error depends on the phylogenetic distance to the reference data, we analyzed the relationship of test and reference genomes in terms of the GTDB taxonomy [17]. In this analysis (Supplementary Section S1), we found that the error slightly increases with the deviation between taxonomic labels. As expected, the MAE for test data was minimal when a reference genome with the same-species label exists. We observed a maximum increase of 1.01%$pt$ for completeness when the taxonomic labels of test and reference data at most agreed up to an order level (see Supplementary Table S3). For contamination, we measured a maximum increase of 1.28%$pt$. Note that this analysis does not reflect how CoCoPyE actually uses the reference data because the prediction always depends on multiple references with possibly varying taxonomy.

Unlike test set A, test set B is not based on a fixed fragment length because it uses the original MAG contigs. In comparison to set A, the results for set B show a slight degradation of the CoCoPyE prediction for completeness with an overall MAE of 4.07%$pt$ (see Table 4). Meanwhile, the contamination error is slightly below the set A results with 3.75%$pt$. The CheckM1 results moved in the same directions, with overall higher values. In contrast, both the completeness error and contamination error slightly decreased for CheckM2 on test set B. However, CoCoPyE still has the best overall prediction performance, yielding the lowest error for completeness as well as for contamination.

The analysis of the signed error distributions shows a similar result like the analysis for set A. Again, CheckM2 errors are clearly biased toward overestimation for completeness and underestimation for contamination (see Fig. 7). Here, CheckM1 shows a more balanced distribution than for set A, but for contamination, the underestimation bias is still visible. Also, for set B, CoCoPyE shows the most balanced distribution for both completeness and contamination error.

For set B, we were able to assign all MAG-based test bins to ecosystem categories based on the main genome component in simulated bins. Therefore, we could analyze the variation of the prediction performance across different habitat types. The results (see Supplementary Tables S4 and S5) show that CoCoPyE has a relatively stable predictive power among all the examined habitat categories, with completeness MAE values ranging from 3.89%$pt$ (engineered) to 4.65%$pt$ (terrestrial) according to a maximum difference of 0.76%$pt$. The contamination MAE values show a maximum difference of 0.45%$pt$ ranging from 3.56%$pt$ (engineered) to 4.01%$pt$ (aquatic). CheckM1 provides slightly more widespread prediction errors among the habitats, with maximum MAE differences of 1.12%$pt$ in completeness and 0.92%$pt$ in contamination. CheckM2 revealed the highest variation between ecosystem categories with a completeness spread of 1.5%$pt$ and a contamination spread of 1.17%$pt$.

## Runtime

In addition to the prediction performance, we also evaluated the runtime and memory usage of the 3 tools. For the evaluation, we used a workstation computer with an Intel Core i9-13900 processor and 64 GB of memory. The tests were run with CoCoPyE 0.2.1 on Debian 12.4. For measuring runtime and peak memory usage, we used the GNU `time` tool. As test sets for runtime measurement, we used 10 mutually exclusive random subsets from our original set of test bins, each containing sequences from 1,000 simulated bins with an average size of 3.8 Mb per bin. While CheckM1 showed the longest average runtime of 94 minutes, 21 seconds, CheckM2 required 63 minutes, 45 seconds and CoCoPyE only 17 minutes, 36 seconds. The runtime variation across different runs was small for all tools. The maximum deviation between 2 different runs was measured for CheckM1 (4 minutes, 30 seconds) and the minimum for CoCoPyE (51 seconds). On average, CoCoPyE required 16.6 GB of memory, CheckM1 36.8 GB, and CheckM2 17.7 GB.

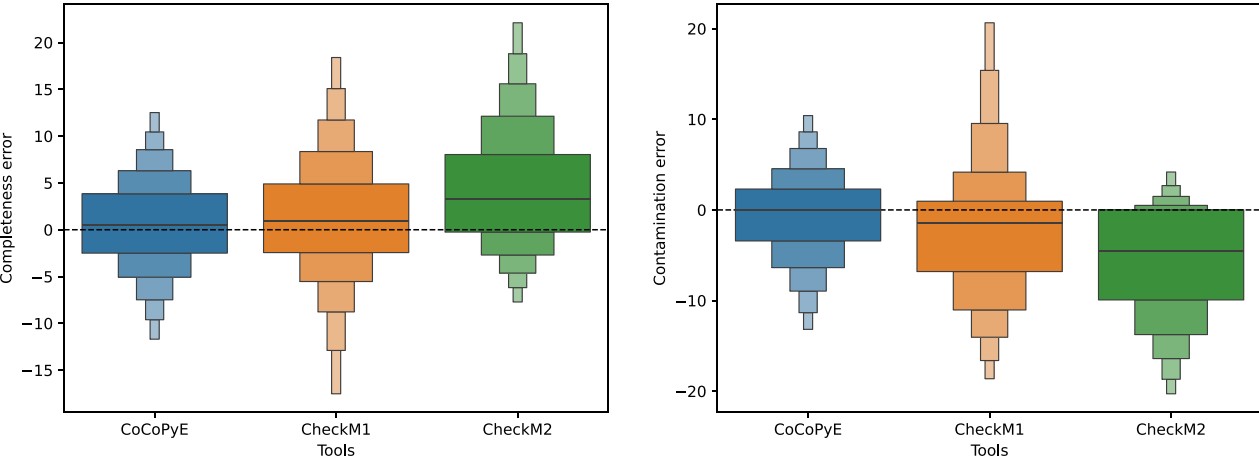

**Figure 7:** Signed error distributions of completeness (left) and contamination (right) for the compared tools on test data set B with errors measured in percentage points. Note that the scale of the y-axis differs between the 2 plots.

## Tool

Our feature-based prediction pipeline is included in CoCoPyE, which is available on PyPI and conda-forge and can be installed with the respective package managers, pip and conda. The source code is available on GitHub (https://github.com/gobics/cocopye) under the GNU General Public License, version 3. The tool supports Windows, Mac OS, and Linux. We also offer a web server (https://cocopye.uni-goettingen.de) to try the tool without installation. In addition to the prediction of genome completeness and contamination, CoCoPyE also provides a lowest common ancestor taxonomic classification for the genome, based on the NCBI taxonomy annotation of the nearest neighbors in the reference database. Furthermore, neighborhood similarities are provided for further analysis of the prediction confidence.

Currently, the user can choose between versions 24 and 28 of the Pfam database as used for the UProC-based feature extraction stage. We do not offer a more recent Pfam database because our results so far indicate that a growing sequence database with an increased number of protein families only has a minor impact on the final prediction performance. In contrast, the impact on RAM storage requirements and startup runtime of UProC is considerable. While UProC with Pfam version 24 runs on 16 GB machines without problems, we recommend 32 GB for version 28, while the actual version 36 would at least require 64 GB of memory. However, with the most recent Pfam version, structure predictions based on AlphaFold [18] have been used to improve many Pfam definitions in terms of more accurate domain boundaries. Therefore, we are currently working on an accessory tool that allows an automatic update to the latest Pfam version if sufficient memory is available.

## Discussion

For several years, CheckM1 has represented the state-of-the-art for predicting genome-quality indices. With CoCoPyE, we now provide a further development of marker-based estimation. The main differences to CheckM1 are the query-specific generation of suitable marker sets and the inclusion of machine learning methods for a refinement of marker-based estimates. In contrast to CheckM1, with CoCoPyE, potential markers are not restricted to single-copy domains. Furthermore, the dynamic marker extraction facilitates the setup and maintenance of the tool because it

overcomes the requirement of a prior definition of lineage-specific marker sets. In CoCoPyE, we use predefined markers only for prior filtering of insufficient input. All markers that are used for prediction are determined at runtime for a particular query genome. Because we do not rely on phylogenetic placement for marker set selection or hidden Markov models for protein domain detection, our tool is also considerably faster than CheckM1. Finally, our results indicate a clear improvement of the prediction accuracy. This can partly be attributed to our 2-stage hybrid architecture. The advantage over purely marker-based tools like CheckM1 or BUSCO is that the second stage in CoCoPyE can learn how to compensate for errors that may result from the restriction to a limited set of markers, particularly with regard to contamination estimates [19].

A direct conceptual comparison with CheckM2 is more difficult, because it is based on a completely different prediction approach and does not use any kind of marker identification. Instead, functional profiles, in terms of frequencies of KEGG orthologs, together with a few other genome content features, are directly used for training and prediction with a machine learning approach. Although the extraction of the relevant information can in principle be learned from such high-dimensional input, the approach requires a large amount of training examples and particular care to avoid overfitting.

Because CheckM2 does not depend on the identification of suitable reference genomes and marker sets, it has the potential to provide reasonable results in cases where CheckM1 does not yield a meaningful prediction [6]. In contrast to CheckM1, CoCoPyE provides a more flexible scheme to evaluate the reference data, but finally, it also depends on the existence of suitable reference genomes. Therefore, CoCoPyE is not intended to replace CheckM2, and possibly a combination of both tools can be beneficial. In particular, if suitable reference genomes for the neighborhood-based analysis exist, CoCoPyE provides a more accurate prediction than CheckM2. Besides the estimation of completeness and contamination, CoCoPyE provides additional information that shows that it is more than just a black box: the number of markers in the stage I prediction, together with the neighborhood similarity scores and the taxonomic classification, may provide an indicator for the confidence of the prediction and may also be helpful to decide in which cases CheckM2 should possibly be preferred for prediction.

## Additional Files

**Supplementary Table S1.** Mean and median absolute error in percentage points for the test set with 50-kb fragments.
**Supplementary Table S2.** Mean and median absolute error in percentage points for the test set with 100-kb fragments.
**Supplementary Table S3.** Mean and median absolute error in percentage points depending on taxonomic closeness of next reference genome.
**Supplementary Table S4.** Mean and median absolute error in percentage points for set B test bins specific to ecosystem categories.
**Supplementary Table S5.** Mean and median absolute error in percentage points for set B test bins assigned to environmental categories.

## Abbreviations

BUSCO: Benchmarking Universal Single-Copy Orthologs; Comp: completeness; Cont: contamination; CRH: count ratio histogram; GEM: "Genomes from Earth's Microbiomes" catalog; GTDB: Genome Taxonomy Database; KEGG: Kyoto Encyclopedia of Genes and Genomes; KNN: K nearest neighbor; MAE: mean absolute error; MAG: metagenome-assembled genome; MIMAG: minimum information about a metagenome-assembled genome; ML: machine learning; MLP: multilayer perceptron; NCBI: National Center for Biotechnology Information; RF: random forest; SCG: single-copy gene; SVM: support vector machine; SVR: support vector regression.

## Availability of Source Code and Requirements

Project name: CoCoPyE
Project homepage: https://github.com/gobics/cocopye
SciCrunch: RRID:SCR_025756
Bio.tools: `biotools:cocopye`
Operating system(s): Windows, MacOS, Linux
Programming language: Python
Other requirements: Python 3.8 or higher, UProC 1.2.0 or higher
License: GNU GPL v3

## Author Contributions

P.M. conceived and supervised the project. N.B., N.L., and P.M. designed, developed, and implemented the tool. N.B. and R.S. implemented the web server. N.L., P.M., N.B., J.S., N.A., and M.B. developed the evaluation setup. N.L., J.S., N.B., and M.B. performed the evaluation. P.M., N.L., N.B., J.S., and M.B. wrote the manuscript.

## Funding

Supported by Deutsche Forschungsgemeinschaft, ME 3138/8-1, P Meinicke.

## Data Availability

An archival copy of the code and supporting data is available via the *GigaScience* repository, GigaDB [14]. The test data that were used for comparative evaluation of the prediction performance are available via Göttingen Research Online [15].

## Competing Interests

The authors declare that they have no competing interests.

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
