## [Peer Review File · GigaScience]

Author's Response To Reviewer Comments

Dear Dr. Zauner,

We are glad and grateful that you are willing to accept our manuscript for publication. In fact, the inclusion of the most recent Pfam version depends on a separate tool (UProC) that showed some severe problems to cope with the increased database size of Pfam 36. So far, we were just able to run a quick code fix on one single computer in our department. All other attempts on different architectures failed and we still do not know the reason for that. We currently try to narrow down the search for the cause by performing numerous tests, but we cannot tell you any date when we will have a satisfying answer. In principle, we could calculate some preliminary results as you suggested, based on the single successful installation with Pfam 36 that we have so far. But we would like to refrain from such a preliminary study because the results will by no means be comparable with the already obtained results with Pfam 24 and 28 for reasons that we explain in detail in our response to reviewer #2 below. For a meaningful comparison we would have to redesign and reimplement our validation framework. We have already started with the collection of redesign ideas, but it could take several months until we have a working solution. Finally, we would like to state again that our tool does not require a more recent Pfam version in order to show a convincing performance. The goal of our tool is not to provide the most accurate Pfam annotation of sequences but to reliably predict genome quality indices.

In the revised version of our manuscript, we now also include the argument of reviewer #2 about the increased quality of Pfam 36 protein domain definitions to underline our current efforts towards the inclusion of more recent Pfam versions (3.3 Tool). In addition, we added a correction of an erroneous statement about the use of single copy domains in CoCoPyE (2.2.1 Reference database).

Best wishes,

on behalf of all co-authors,

Peter Meinicke

Dear Reviewer,

we appreciate your interest in the inclusion of the latest Pfam version into the CoCoPyE tool and its evaluation, and we also see the potential benefits of such an inclusion. Therefore, we are grateful that you indicated the advantages of the most recent Pfam version, and we have included the information about the AlphaFold-related improvements of protein domain definitions into the revised version of the manuscript (see section 3.3).

However, although it may be technically possible to provide some preliminary results for Pfam 36 we have some serious reasons against such an evaluation.

First of all, we still do not have a general fix of the UProC related problem with the increased Pfam database size. It is actually some sort of rapid fix that we applied and that so far runs on one computer in our department. For reasons that we do not know other attempts on different architectures failed so far. In our opinion, this is currently not a sufficient basis for the publication of preliminary results.

Secondly, even if we could provide a general solution for the UProC-related problem at the moment, the evaluation of the prediction performance would not provide us with results that are comparable with the already achieved results based on Pfam 24 and 28. As explained in the manuscript, we have spent much effort to implement a special validation pipeline, that is able to adjust all (hyper) parameters of our prediction engine automatically. The validation is based on a date splitting scheme on the reference genomes that tries to minimize the overlap between training and validation data while providing a realistic

split that mimics an update with more recent genomic data. In this context, the older Pfam versions that we use warrant that the UProC engine never violates the date splits. Therefore, using the same validation scheme with Pfam 36 would not yield results that are directly comparable with the existing results due to an increased overlap of genome-related data between training, validation and test sets. In fact, the inclusion of more recent Pfam versions would require the redesign of our validation pipeline and possibly also a refined test setup. We have already begun with some research into this direction but also in this regard we do not have a working solution now.

Thirdly, the goal of CoCoPyE is not to provide an accurate Pfam annotation of sequence data but rather to predict genome quality indices. Pfam frequency profiles are used as feature vectors which we consider as "noisy" due to inevitable domain annotation and detection errors. The subsequent stages of our prediction engine do not rely on noise-free features and to some degree can cope with errors that happen in the first stage. If we would really insist on features with minimal noise, we would have to use HMMER for the Pfam detection stage. For reasons of computational speed, we do not see this as a realistic option. Besides that, we would like to point out, that a more recent Pfam version is not necessary to show the superior CoCoPyE prediction performance.

We hope that you can understand our reasons and see the efforts we are making to find a truly satisfactory solution for the inclusion of the latest Pfam version.

Best regards,

on behalf of all co-authors,

Peter Meinicke